# The Path from Traditional Fisheries to Ecotourism in Cimei Island

**Wei-Ying Sung [1], Hsiao-Chien Lee [2] and Wen-Hong Liu [2,*]**

1    Department of Sports and Leisure, Hungkuang University, Taichung 433304, Taiwan
2    Institute of Marine Affairs and Business Management, National Kaohsiung University of Science and Technology, Kaohsiung 811213, Taiwan
*    Correspondence: andersonliu@nkust.edu.tw; Tel.: +886-7-3617141 (ext. 23131)

**Abstract:** Cimei Island is a second-class outlying island. In recent years, due to the lack of coastal fishery resources and restrictions on traffic and climate, the traditional fishery and tourism industries that residents rely on for a living have faced challenges. This research is based on the Barbados Programme of Action, from the perspective of environmental conservation, industrial economy, and social development in sustainable development, and from the perspective of local stakeholders, to construct a sustainable tourism action approach and development mechanism in Cimei Island. A qualitative research method was adopted. Various sources of data, including focus discussions and in-depth interviews with local stakeholders, textual materials, and field observations, were collected and analyzed. The results are as follows: (1) Cimei faces great threats in social, economic, and environmental aspects, which has led to Cimei Islands' promotion of permanent development. (2) The current development of Cimei cannot effectively drive industrial development or the transformation of fishing villages. However, residents are worried that overdevelopment will impact the ecological environment and lifestyle on the Islands. (3) Cimei's sustainable tourism development approach should be to first take inventory of the ecological environment and cultural resources, plan related environmental laws and regulations, and finally, use sustainable tourism to drive industrial development to carry out island development.

**Keywords:** sustainable tourism; island development; Cimei Islands; local stakeholders; Barbados Programme of Action

## 1. Introduction

### 1.1. Background and Motivation

The Cimei Township is in the southern region of the Penghu Islands; it is rich in geographical environment resources, marine resources, and fishery resources [1]. Cimei Island is part of the outlying islands of Taiwan (second-level outlying islands). Similar to the development difficulties of most islands, the fishing industry that Cimei Island relies on has been drastically reduced due to overfishing in recent years. In addition, the aging population and outflow of young adults have led to the aging of the industrial population in fishing villages. In terms of tourism development, due to the lack of in-depth sightseeing itinerary planning in Cimei, most tourists choose short-distance and island-hopping tours, which can only bring limited tourism income to the Island [2–4]. As a result, it is essential that an integrated plan is devised for the development of Cimei Island.

Islands are inherently vulnerable to climate change, so when planning for island development, the blue economy concept of economic integration of ocean, coastal, and fishery resources should be considered [5,6]. Previous studies have pointed out that Cimei lacks a systematic plan for sustainable island development; coupled with the decline of fisheries and shallow-dish sightseeing, it is difficult to carry out sustainable development and industrial transformation of Cimei Islands [4,7,8]. In other words, it is important to

construct a sustainable development model for islands by considering their fragility and the need for industrial transformation.

*1.2. Purpose*

Past studies have pointed out that if Cimei Island can plan island development based on environmental sustainability, and integrate environmental resources and cultural features to develop ecotourism, it will help the transformation of the Island's industry and the sustainable development of its fishing villages, and it could eventually become a model island for sustainable tourism ecology [4,8,9]. According to previous studies, most of the sustainable research on Cimei Islands has employed a top-down view of government governance; the lack of a bottom-up view from the stakeholder position makes it difficult to reach a consensus on the planned development direction and follow-up development. For this reason, this research aims to gradually construct the direction and strategy of sustainable tourism action on Cimei Island from the perspective of sustainable tourism development through the perspectives of local stakeholders. Based on the results, this study presents policy recommendations and suggestions for further research.

Based on the research background and motivations, the research purposes are as follows:

(1)  Inventory the challenges of sustainable tourism development in Cimei Island.
(2)  Summarize the views and attitudes of local stakeholders on the sustainable tourism development of Cimei Island.
(3)  Propose the sustainable tourism development strategy and action path of Cimei Island.

## 2. Literature Review

*2.1. Implications of Practical Sustainable Development of Islands*

In 1992, the United Nations Conference on Environment and Development (UNCED) held an Earth Summit in Rio to jointly seek "sustainable development", i.e., development that meets the needs of the present without compromising the ability of future generations to meet their own needs. Among them, "Agenda 21" called on countries, regions, and local governments to take action to establish global partnership and achieve sustainable development. The 17th chapter of Agenda 21 focuses on marine protection, including enclosed and semi-enclosed oceans, as well as the protection, rational use and development of coastal areas and their biological resources. Under this goal, preventive and predictive management programs were developed, including the sustainable development of small islands.

Small islands are rich in marine ecological resources, with a high degree of biodiversity. Their locations are also of significant importance in terms of national defense strategy. However, due to their geographical dispersion, limited resources, isolation from major markets, and ecological fragility, such as the impact of climate change, economic development is not easy, and it is necessary to avoid the development of economies of scale [10]. Based on these conditions, Agenda 21 proposes two major goals, three major actions, and four major practical methods for the sustainable development of small islands (Table 1). Management plans and promotion mechanisms seek to establish suitable paths for island sustainable development.

In order to implement the island sustainable development action plan mentioned in Agenda 21, the Declaration of Barbados, and the Programme of Action for the Sustainability Development of Small Island Developing States, referred to as Barbados' Programme of Action, is of primary importance. The Barbados Programme of Action for the Sustainable Development of SIDS (BPOA), a 14-point program that identifies priority areas and specific actions necessary for addressing the particular challenges SIDS faces, such as climate change and sea level rise, natural and environmental disasters, waste disposal, coastal and marine resources, fresh water supply, land resources, energy, tourism and recreation, biodiversity, government mechanisms, regional institution and scientific and technological cooperation,

transportation and communications, science and technology, talents and education, policy supervision, and other topics [11].

**Table 1.** Programme of Action for sustainable development of small islands in Agenda 21.

| Goals | Actions | Practice Methods |
|---|---|---|
| 1. Use marine and coastal resources to satisfy Island residents, improve the quality of life, and maintain biodiversity. 2. Develop effective and creative local programs to reduce ocean changes and threats to coastal environmental resources. | 1. Develop management plans, such as inventory of ecological, economic, and social resources of small islands; formulate and monitor environmental carrying capacity; plan site selection and environmental assessment according to the characteristics of small islands; introduce community participation in the planning process; inventory existing policies and practices; respond to the impact of climate change based on prevention methods; promote the development of environmentally friendly technologies. 2. Establish the ecological, economic, and social data on a small island, and use the geographic information system to plan the development suitable for the characteristics of a small island. 3. International and regional cooperation and coordination. | 1. Financial and cost evaluation. 2. Develop scientific and technological methods to monitor marine resources. 3. Talent training and capacity building. |

Source: United Nations (1994). Programme of Action for the Sustainable Development of Small Island Developing States.

Before the Barbados action plan is promoted, it is necessary to investigate the Island's ecological environment, industrial economy, and social and cultural aspects. At the same time, the existing policies and practices should be counted, community participation and local stakeholders' opinions should be introduced, and sustainable development paths formulated from them. Certain mechanisms can effectively promote the sustainable development of islands. In other words, if you want to carry out sustainable island development, you must consider the balanced development of environmental conservation, social equity, and economic growth at the same time.

*2.2. Concept for Sustainable Tourism Development of Islands*

Penghu County is rich in natural environmental and ecological resources, with diversified precious resources such as sea and land ecology, topography and geology, and human landscapes [12]. However, it is also faced with the challenges of local industrial development and the maintenance of residents' quality of life. How to protect natural and humanistic environmental resources, while revitalizing the economy and maintaining social stability, thereby achieving sustainable development, requires overall high-level planning [5], such as fostering sustainable development framework indicators. The formulation of ecological conservation methods, the establishment of island carrying capacity and landing fees and other standard methods are the directions for promoting the sustainable development of islands [13,14]. Liu et al. conducted a study on the local revitalization and development of Cimei Island, and found that the relevant income generated by the promotion of local tourism industry is low, and there is a lack of clear environmental ecological conservation strategies and environmental carrying capacity-related regulations [9]. The local revitalization concept is hoping to put together the local and industry talents to boost local development, strengthen local culture, and exhibit the local beauty [15]. To promote local development, it is necessary to take an inventory of environmental resources and clarify the local consensus, to promote the transformation and development of island industries.

"Sustainable Tourism" is one possible way to promote the transformation of island industry and local development. Sustainable tourism needs to explore the most suitable type of tourism for an industrial economy, ecological environment, and social development

from the perspectives of environmental, economic, and social sustainable development [16], while maintaining the value of the natural ecological environment system and considering the economic and general well-being of residents [17,18]. Multiple stakeholders on the island had different views on tourism development, especially when discussing from different aspects, such as environment, economic, and social development; it has been found that stakeholders pay attention to different aspects [19]. If the tourism development process can effectively link the original "people, culture, land, production, and scenery" characteristics of the area, it will not only increase the income of residents, but also provide jobs in traditional industries, that is, it will not only reduce the outflow of people from agricultural and fishing villages, but also solve the problems of island economic development and industrial transformation [9,17].

In general, it can be seen from previous studies that it is necessary to consider environmental and ecological sustainability. Under the conditions of lifestyle and social development without adverse impacts, an action plan can be generated through the establishment of a consensus among local stakeholders to derive the opportunity to introduce the concept of sustainable tourism, as well as promote industrial transformation and local development.

## 3. Methodology

This research refers to the goals, actions, and practical methods of Barbados' Programme of Action. We investigated the ecological environment, industrial economy, and the humanities and society aspects of the Island, and took stock of existing local policies and practices. Through focus discussions and in-depth interviews, we introduced community participation and local stakeholders' opinions, and formulated a sustainable development path for Cimei Island. To effectively understand the views of local stakeholders, we further used textual materials combined with field observations, in-depth interviews, and expert focus seminars to collect various materials to summarize and understand the views of local stakeholders concerning the sustainable tourism development strategies and action paths of Cimei Island.

### 3.1. Scope and Subjects

This study uses Cimei Island (Township) as the research field, and applies residents, marine recreational operators, government representatives, and participating scholars as stakeholders. Cimei Township is located at the southernmost tip of Penghu County, adjacent to the four southern islands. It has a rich geographical environment and marine resources. The Island has a total area of about 6.99 km$^2$ and a coastline of 14.40 km (see Figure 1). It is the fifth-largest island in the Penghu Archipelago. There are six villages on the Island: Donghu Village, Xihu Village, Zhonghe Village, Pinghe Village, Haifeng Village, and Nangang Village. The total population is 3901, and the total number of households is 1433. Fishery and tourism are the main industries of the Island [20–22].

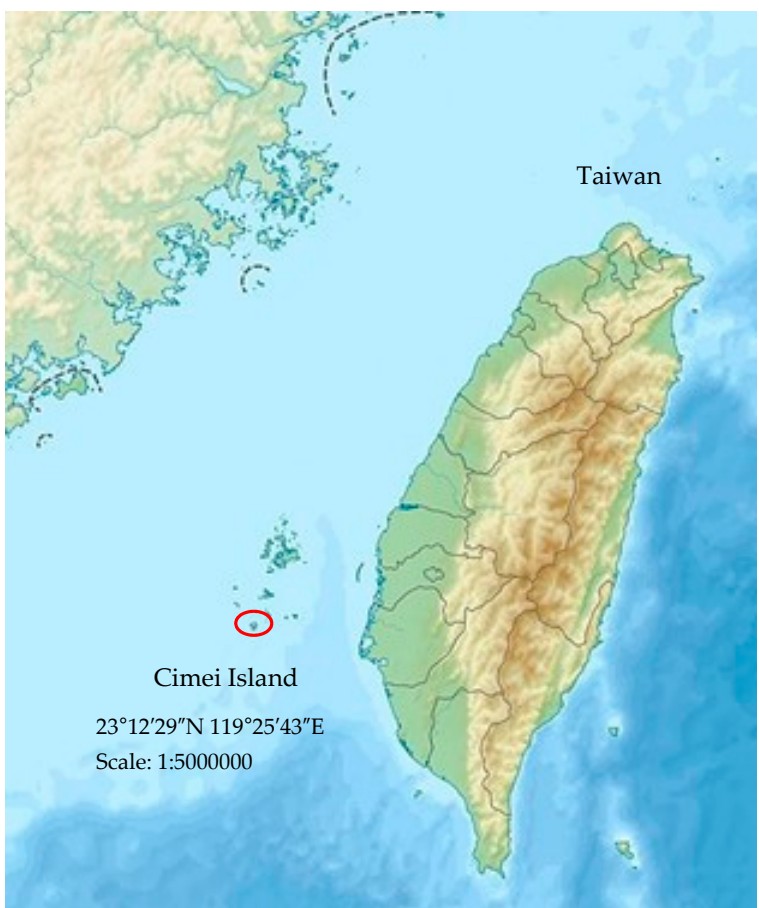

**Figure 1.** Geographical location map of Cimei Island.

*3.2. Qualitative Data Collection*

The research steps are divided into four parts: textual analysis, participatory observations, in-depth interview, and expert focus symposium, Firstly, we collected textual materials related to Cimei, such as Cimei Township Records, local development articles, and reports. Secondly, we went to Cimei to observe the relationship between merchants, vendors, local groups, and residents in important settlements such as Nanhu Port, Twin-Hearts Stone Weir, and Fishery Resource Reserve, as well as to further reflect on the relationship between environmental ecology, industrial development, social connection, and island development.

Thirdly, semi-structured interviews were conducted with individuals among the local stakeholders. The semi-structured interviews allowed the interviewees some flexibility in answering and clarifying questions. These interviews were carried out on a one-to-one basis, lasting for no more than 30 min each. The interviews were conducted face-to-face so that non-verbal communication could be observed during the interviews. A total of 13 individuals were interviewed. Among them, three were the Cimei Township leaders, such as the township mayor, the village head, and the village representative, three were the tourism industry workers, four were residents and fishermen, and three were scholars and experts. Their consent to participate in the study was also obtained.

The interview questions were as follows:

1. Impressions and opinions regarding Cimei's past and present?
2. What are your expectations and opinions concerning Cimei's future vision?
3. What do you think about Cimei's development of sustainable tourism?
4. Suggestions for Cimei to develop sustainable tourism.

Finally, we also organized three focus groups for scholars, experts, representatives of local organizations, and residents to understand the opinions of local stakeholders on Cimei's promotion of sustainable tourism development.

### 3.3. Content Analysis

This study summarizes the data relevant to the text; highlights the observations, interviews, and meeting processes; and uses the triangulation method to cross-compare different types of data to reduce research bias [23]. In addition, the participants were provided with the collected data to review, and if there were any disagreements, they would communicate and discuss them until opinions reached a consensus to ensure the consistency of the data content [24].

## 4. Results and Discussion

### 4.1. Challenges of Sustainable Tourism Development

From the perspectives of environmental conservation, industrial development, and social development, this study combined past relevant data and field surveys to summarize the sustainable development dilemmas of Cimei Island, as follows.

#### 4.1.1. Challenges in Environmental Conservation

This study found that Cimei Island may encounter dilemmas in environmental conservation and sustainable development (Table 2), as the current environmental protection policy is unknown, and the garbage and recreational impact caused by short-term tourists may not only damage the Island's environmental ecology, but also impact the lives of residents on the Island. In addition, due to the lack of clear coastal biological resource surveys and a complete monitoring mechanism in the protected area, although the Island has a fishery resource protection zone and rich coastal ecological resources, it is unable to provide effective supervision due to the lack of human resources and systems. It is therefore not easy to further plan for in-depth sightseeing and recreation.

**Table 2.** The dilemma of island environmental conservation.

| Dilemma | Examples |
| --- | --- |
| Environmental protection strategy is unclear | During the peak tourist season, tourists create a lot of garbage, which endangers the marine ecology of Cimei, and there is no effective marine ecological protection strategy for the Island. |
| Lack of coastal biodistribution surveys | The intertidal zone is rich in ecology, but there is a lack of ecological statistics and distribution surveys, and it is impossible to effectively conserve and manage the intertidal zone organisms. |
| Lack of perfect supervision mechanism in fishery protected areas | There are fishery resource reserves and no-fishing areas on the Island, but due to the lack of manpower, resources, and systems, effective supervision cannot be provided. |

This result is similar to that of Huang and Chen [8]. That is, Cimei's overall recreational resources need to be further systematically planned. In terms of Island area and environmental carrying capacity, reasonable control must be maintained for the short-term influx of the tourist population. Some studies have further pointed out that island environmental conservation needs to establish clear tourism development norms and regulatory mechanisms. [4,23].

#### 4.1.2. Challenges in an Industrial Economy

This study found that islands may encounter difficulties in promoting sustainable development of the industrial economy (Table 3). Traditional fisheries are diminished due to the aging population and the outflow of young and middle-aged people. In addition, the coastal fishery resources are facing exhaustion, resulting in the need for industry. Development is facing difficulties, and the tourism industry that it relies on is based on

a single-point, island-hopping itinerary plan. Most tourists come to the Island for short stays, and it is difficult to produce in-depth and large tourism benefits. In addition, the Island lacks clear industrial development regulations, such as island carrying capacity and tourism industry development-related regulations. Therefore, it is also difficult to promote the transformation and development of Cimei's industry.

**Table 3.** The dilemma of island industrial economy.

| Dilemma | Examples |
| --- | --- |
| Fishery industry faces fault and decline | The catastrophe of coastal fishery resources, the high transportation cost of fishery products, and the aging of the industrial population, cause industrial development to fall into decline. |
| Monotonous journey and lack of in-depth planning | Most tourists travel for half a day. For the island, residents can only earn a small amount of transportation and meal expenses, which has limited benefits for the development of island tourism. |
| Sea-based recreational activities are too monotonous | Sea-based recreational activities are mainly snorkeling, other sea-based recreational activities such as SUP, canoeing, diving, etc., have been properly planned, and the abundant marine resources have not been fully utilized to increase the income of related industries. |
| Lack of normative measures for the development of tourism industry | Residents have expectations for the development of the tourism industry but worry about the ecological impact. In addition, due to the lack of research on environmental ecological carrying capacity in the past, there is no scientific evidence to establish regulations for the development of sustainable tourism industry. |

This result is similar to that of Zhang [3] and Huang and Chen [8], both of which pointed out that Cimei has inherent structural disadvantages in tourism development. Due to the long travel time to Cimei and the lack of in-depth sightseeing itinerary planning, most tourists choose half-day to one-day trips around the Island, and the benefits to the Island are limited; most are concentrated in the catering industry around Nan-hu Port and shops around the famous scenic spot "Twin-Hearts Stone Weir".

### 4.1.3. Challenges in Humanities and Society

This study found that the Island's possible dilemma in promoting sustainable development in terms of humanities and society at this stage includes population structural problems such as the aging population and the serious outflow of young and middle-aged people, resulting in a lack of momentum and support on the Island (Table 4). The cultural landscape is systematically counted and preserved; in addition, because it belongs to traditional fishing villages, it is relatively conservative in local culture and concepts, and is highly exclusive.

**Table 4.** The dilemma of island humanities and society.

| Dilemma | Examples |
| --- | --- |
| Aging and serious outflow of young adults | Due to the lack of employment and development opportunities, the outflow of young and middle-aged people, and the serious aging of the Island itself, the Island lacks development momentum. |
| Lack of preservation of cultural landscape | It has unique local culture, such as stone pagodas, stone tablets, stone gandangs, and temples, but lacks complete investigation and preservation. |
| Conservative and xenophobic local ideas | The Island is small and has a small resident population. The residents are strongly connected to each other, but their concepts are relatively conservative, and they are relatively cautious about external factors. |

This finding is similar to that of Wang and Xu [25] that is to say, the society and human resources of the small island are closely co-constructed with the island environment, and

the residents had strong family and social relationships. In addition, it is also similar that the aging society leading to the gradual decline and depression of industry.

In general, the main problems that Cimei faces cover environmental ecology, industrial economy, and social development. In regard to environmental ecology, due to the lack of clear ecological environment conservation strategies and environmental carrying capacity specifications, environmental protection efficiency is low. In addition, residents worry that their quality of life will be impacted after opening the Island for sightseeing, and they will be unable to maintain the existing style of the fishing village. In the economic facet, traditional fisheries are depleted of fishery resources and experience high fishing costs, and the younger generation are unwilling to invest, which makes fishery development difficult. Meanwhile, the tourism industry focuses on single-point and island-hopping tourism, which can bring limited tourism benefits and is not relevant to the local area. In terms of social development, Cimei is a small island with a small population size. The Island has a permanent population of about 2000. The island has strong social connections, but is relatively xenophobic and conservative in regard to the concept of sustainable tourism development.

### 4.2. Views of Stakeholders on the Development of Sustainable Tourism

This research found that although Cimei Island faces challenges in environmental ecology, industrial economy, and social development, local stakeholders have expectations for Cimei's industrial transformation and local development. As mentioned by the mayor, residents, and business owners, it is necessary to devise an advanced environmental conservation plan to ensure that the existing ecosystem and quality of life can be maintained before it is possible to introduce sustainable tourism development.

> We hope to have the opportunity to develop tourism, but we are also worried that Cimei will be like Liuqiu. We are worried that if tourism develops without planning, the overladed tourist will destroy the environment and ecology on which we depend . . . *Township Mayor 20 August 2020.*

> I was not used to staying in a big city before...I still used to come back to Cimei. I prefer the slow pace of the fishing village. I hope to promote local employment and tourism, but I am afraid of losing its original flavor.... Environment and development are difficult to balance. We hope that they (tourists) will earnestly experience Cimei. Fun should be to earnestly experience local life . . . *Resident Miss Chen 26 December 2020.*

> In the past, the most common problem we observed when conducting industrial transformation counseling in traditional agricultural and fishing villages was the aging of the industrial population in fishing villages and the outflow of young and middle-aged people, causing the development of fishing villages to wither day by day. Relevant government agencies are concerned about the development of fishing villages and the transformation of islands, hoping to provide better development opportunities for the next generation, and to allow fishing villages to create sustainable development while protecting fishery resources and the ecological environment. *Scholars 14 September 2020.*

In general, residents hope to find possible development paths for the Island's traditional fishery development and island tourism development models, but they are also worried about the destruction and impact of existing marine resources and the Island's ecological environment caused by the excessive development of Liuqiu and Green Island. This result is similar to the findings of Huang and Chen [8]. As a matter of fact, Cimei Island can seize the opportunity for sustainable development to introduce the concept of the Caspian Sea, focus on fishery resource conservation and local cultural heritage, develop in-depth tourism, and create local employment opportunities. As for the direction of the transformation of a sustainable ecological island, most stakeholders on the Island expect that it can start from the perspective of environmental and ecological conservation,

while preserving the ecological environment and the cultural context of the fishing village, considering the development of an industrial economy and quality of life.

*4.3. Constructing a Sustainable Tourism Development Strategy*

Sustainable development needs to consider three major elements: environment, economy, and social. Firstly, in the process of economic development, it is impossible to maintain true sustainability without environmental and social support. Secondly, without economic support, sustainable social development and environmental ecology cannot be pursued. Lastly, without the support of a stable social system, environmental ecology and economic development will not last.

The same is true for sustainable tourism development, which must consider environmental conservation, tourism industry growth, and socioeconomic development. Among them, "environmental sustainability" is the key to determining whether the needs of tourists can be met, and whether industry can grow effectively.

> If we can first conduct an inventory survey of Cimei's ecological environment carrying capacity, marine spatial planning, marine resources, and fishing village culture, and use the results to plan Cimei's relevant development standards in advance, and then open tourism development and industrial transformation, it will be possible to achieve a balance between environmental resources and industrial development, and consider island development and the livelihood of islanders . . . *Employer Mr. Wang 30 September 2020.*

> The most difficult thing to create and develop an island is to communicate with ideas. What is most to fear is the established impression of the people. If you are unwilling to come in contact, there will naturally be no opportunity to change the locals' impression. Our role is to help create this need and possibility with our experience in Liuqiu. *Director of Undersea Players 27 December 2020.*

> Tourism development is not about a balance between failure and environmental conservation; it must be promoted in the context of resource conservation so that a balance can be found between development and the environment . . . *Scholar 12 April 2020.*

How to effectively use and maintain natural resources and make the precious ecological environment sustainable is one of the important issues. Cimei Island needs a sustainable tourism development approach, as mentioned by Yan [14]. To achieve this, it must first be constructed for evaluation. The implementation of sustainable development framework indicators, the establishment of protected areas, and the formulation of relevant laws and regulations on ecological conservation, etc., are the long-term goals of the relevant management units, and it is possible to further protect the environment through sustainable tourism development paths, industrial transformation, and local development.

The initial consensus of the local stakeholders on sustainable tourism, in-depth tourism, and the creation and development of fishing villages is that when a fishing village is transformed, it is necessary to conduct basic environmental resource surveys, such as fishing village cultural preservation, marine resource survey, and marine spatial planning and use. Based on "sustainable tourism" and "good cooperation", island creation and fishing village transformation and development were carried out. The development direction of this sustainable tourism guidance industry not only conforms to the sustainable development elements, such as "environmental protection", "economic development", and "social equity", but also recognizes that the protection of environmental resources is the most important development and start-up key among them.

*4.4. Policy Suggestions*

From the above research results, Cimei Island has its difficulties in environmental ecology, industrial economy, and social development. However, from the investigation process, the local stakeholders expect to promote the transformation of the Island's in-

dustry through the concept of sustainable tourism. It is suggested that we can refer to the sustainable development experience of Sipadan Island in Malaysia [10,13,26], through the collection of relevant fees and the total amount control method, so that the ecological conservation and tourism development can be based on price, reduce the environmental burden, improve the quality of recreation, and finally, achieve the goal of ecotourism.

The sustainable tourism-driven model of islands can be carried out in three stages: resource inventory, consensus building, and cooperative execution. First, through the inventory of geographical environmental resources, marine environmental resources, and cultural resources of fishing villages, we can understand the sustainable tourism development resources of Cimei Island, and build a sustainable environmental carrying capacity. Second, we can build a consensus and generate an action plan through local stakeholder development meetings. Finally, we can implement an action plan with fishermen and residents to promote the industrial transformation and local development of Cimei Island.

*4.5. Suggestions for Follow-Up Research*

This research is based on textual materials, participatory observations, and in-depth interviews to understand the views of local stakeholders concerning the sustainable tourism development of Cimei Island and recommendations for promotion. It was initially discovered that under the premise of resource sustainability, the Island is willing to drive its industrial transformation and local development through the concept of sustainable tourism.

Nevertheless, the opinions of the Islanders cannot be fully represented by the viewpoints of the local stakeholders. The opinions of its residents need to be further investigated by follow-up research. Therefore, this research suggests that a questionnaire should be compiled based on the concept of sustainable tourism development to test whether residents understand its feasibility.

## 5. Conclusions and Suggestions

The crux of the sustainable tourism development of Cimei lies in the transition process of the tourism and fishery industry. Therefore, it is necessary to consider the residents' lifestyles, industry, and ecological environment when seeking to make it a complete system. From an industrial perspective, most residents rely on fisheries and tourism for their livelihoods, yet fisheries have been facing the dilemma of reduced catches in recent years. In the tourism industry, although the current island tourism model can bring benefits to local hotels, restaurants, and water activities businesses, it is not yet at an industrial scale. In other words, the current tourism development model cannot effectively drive the transformation and development of Cimei's industry. However, residents are also worried that the concept of sustainable tourism and the development of islands will impact the original island lifestyle. Therefore, although local stakeholders have expectations, most of them also have reservations.

From the perspective of a development strategy, local stakeholders hope to take inventory of Cimei's ecological environment carrying capacity, marine spatial planning, marine resources, and fishing village culture, and devise a clear inventory of the Island's ecological environment, cultural history, marine conservation, and other resources, and establish relevant environmental development. After standardization, tourism development and industrial transformation are gradually being carried out. In general, if Cimei Island can be based on environmental conservation and sustainable development, and can establish environmental carrying-capacity-related legal regulations, then island development planning and promotion can be carried out. Only in this way can we consider the sustainable development of the Island in terms of industrial economy, social development, and environmental sustainability.

**Author Contributions:** This study has three authors. W.-Y.S. and W.-H.L. are mainly responsible for theoretical framework, conceptualization, and writing, while H.-C.L. is responsible for investigation and administration. All authors have read and agreed to the published version of the manuscript.

**Funding:** This research was funded by Ocean Affairs Council, Taiwan, grant number 1100611-DBCD.

**Institutional Review Board Statement:** This study excludes this statement as it does not involve human or animal studies.

**Acknowledgments:** In this study, we would like to thank the reviewers for their comments and suggestions on this paper, as well as the Ocean Affairs Council, Penghu County Government Agriculture and Fisheries Bureau, and Cimei Township Office for their assistance in making this study a success.

**Conflicts of Interest:** The authors declare no conflict of interest.

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
