# Peer review of "The Path from Traditional Fisheries to Ecotourism in Cimei Island"

_fishes, doi:10.3390/fishes7040200_

Round 1

Reviewer 1 Report

Line 176 typo in secondly

line 186 change are to were

Results and discussion should be separate.

The results do not specify the following:

Number of people participating in the interviews. 

Number of people that declined to participate

The number of experts contributing to the expert focus symposium

A better analyses of the responses should be included rather than pasting comments from the respondents.

Section 5.2 should be expanded further

Author Response

Reviewer’s Comments

Author Responses

Reviewer #1

Line 176 typo in secondly

This sentence has been revised.

line 186 change are to were Results and discussion should be separate.

Results and discussion have been separated.

The results do not specify the following:

Number of people participating in the interviews.

Number of people that declined to participate

The number of experts contributing to the expert focus symposium

The number of people who have supplemented the interview and the information of the experts who participated in the expert focus symposium.

A better analyses of the responses should be included rather than pasting comments from the respondents.

The analysis has been supplemented in the text and compared with other island development cases, and then the viewpoints of this paper are summarized.

Section 5.2 should be expanded further

Section 5.2 has been expended

Reviewer 2 Report

I am not sure that this journal is appropriate for such an article. Fish and their environment are peripheral threads. This is an article considering theoretical issues of socio-economic phenomena. Numeration of cited literature in the text needs to be ordered. Figure 1 - legend and geographical names should be in English. A scientific discussion would need to be developed. Conclusions should be completed after extended scientific discussion.

Author Response

Reviewer’s Comments

Author Responses

Reviewer #2

I am not sure that this journal is appropriate for such an article. Fish and their environment are peripheral threads. This is an article considering theoretical issues of socio-economic phenomena. Numeration of cited literature in the text needs to be ordered.

References cited in the text have been ordered

Figure 1 - legend and geographical names should be in English.

Legend and geographical names have been revised in English in Figure 1.

A scientific discussion would need to be developed.

Supplementary and expanded scientific discussions already included in the article.

Conclusions should be completed after extended scientific discussion

The extended scientific discussion has been supplemented in the article.

Reviewer 3 Report

Dear Authors

I appreciate your effort on this research and intend of sharing the knowledge with the public and the scientific community. It is important to have sustainable development approaches in development activities based on proper scientific basis. Your effort and findings will be supported to the future decision-making in sustainable ecotourism particularly in Cimei Island and other similar places in the region and globally.

Your manuscript is interesting and strengthen the knowledge base of sustainable development approaches. However, as a reviewer I have seen some possible improvements in some sections of your manuscript. I believe that authors will be able to address those comments in the improved version.

Introduction

1.       There is a big gap between first two paragraphs of the introduction, which damages the writing flow. Better to include few sentences to create the link between these two paragraphs.

2.       The literature review is depend on very few publications, which the information are concentrated on "Agenda 21" and few local literature.  It is better to give information on wider knowledge about the global concerns in similar type of studies (i.e. concerns, issues, obstacles, when introducing sustainable development planning etc.)

Figure 1.

1.       It is important to indicate the location of the study site (Island) interrelation to the main state

2.       Change the language no comments can give without understanding the labels.

3.2 Data collection

Need to elaborate the sample. You may include a table indicating the number of personals interviewed in different categories. It will be more appropriate if authors can indicate the special distribution of the samples in the island (i.e. village-wise). Indicate the basis for selecting the individuals for the sample.  Explain why the said groups were selected as stakeholders

Results and discussion

It is more appropriate to take evidences from studies in other parts of the world/region. For an example authors may get the support of the similar kind of studies conducted in other parts of the world and improve the discussion. Some of the factors influencing on this type of development activities may defferent from the present observations and will be useful to go for better recommendations. Therefore, it is better take the support of published literature in the discussion.

Literature review

Follow standards of reference writing. Some references are starting from initials some are starting with last name, some references have provided the page numbers some are not. Need to improve the reference list.

Some other comments and suggestions are given in the manuscript for your consideration

Thank you

Author Response

Reviewer’s Comments

Author Responses

Reviewer #3

Introduction

1.  There is a big gap between first two paragraphs of the introduction, which damages the writing flow. Better to include few sentences to create the link between these two paragraphs.

One sentence has been added to strengthen the connection between the first and second paragraphs, as is presented in the following.

As a result, it is essential to come up with an integrated plan for the development of Cimei Island.

2. The literature review is depend on very few publications, which the information are concentrated on "Agenda 21" and few local literature.  It is better to give information on wider knowledge about the global concerns in similar type of studies (i.e. concerns, issues, obstacles, when introducing sustainable development planning etc.)

The island development and similar studies have been supplemented in literature review.

Figure 1.

1. It is important to indicate the location of the study site (Island) interrelation to the main state

2. Change the language no comments can give without understanding the labels.

The Figure 1 has been revised as follow.

1. It is indicated the location of the study site (Island) interrelation to the main state

2.It has been marked in English in Figure 1

3.2 Data collection

Need to elaborate the sample. You may include a table indicating the number of personals interviewed in different categories. It will be more appropriate if authors can indicate the special distribution of the samples in the island (i.e. village-wise). Indicate the basis for selecting the individuals for the sample.  Explain why the said groups were selected as stakeholders

The information of the interviewee has been supplemented in the text.

Results and discussion

It is more appropriate to take evidences from studies in other parts of the world/region. For an example authors may get the support of the similar kind of studies conducted in other parts of the world and improve the discussion. Some of the factors influencing on this type of development activities may defferent from the present observations and will be useful to go for better recommendations. Therefore, it is better take the support of published literature in the discussion.

Results and discussion have been supplemented in the text and compared with other island development cases, and then the viewpoints of this paper are summarized.

Literature review

Follow standards of reference writing. Some references are starting from initials some are starting with last name, some references have provided the page numbers some are not. Need to improve the reference list.

The references list has been revised.

Reviewer 4 Report

I would like to thank the authors for conducting such an interesting study. The title is attractive and appropriate. The introduction, results and discussion are well organized and well written in a rhythmic way. However, I still have some concerns which I describe below:

The abstract of this manuscript is not well organized. The authors did not mention their methods of data collection. I would like to see a short description of the methodology, results, and recommendations

Line 39: This sentence is not clear.

Line 85: Rewrite the sentence for better clarity.

Line 97 – 98: What is "Barbados' Programmer of Action"?

Line 99: The Barbados Programme of Action for the Sustainable Development of SIDS (BPOA), a 14-point programme that identifies priority areas and specific actions necessary for addressing the particular challenges SIDS faces. Therefore, correct the sentence.

Line 105: No reference.

Line 124: What is "local creation"?

Line 170: Is this article only for the Chinese people? If not, change the texts on the map. The resolution of the map is also very poor. My suggestion is to use a different map. Unfortunately, the readers won't be able to find the exact location of the sampling site on this map.

Line 172: Section 3.2 needs a radical change. This section lack clarity and completeness. The authors should not mention "various" but rather mention the specific name of the materials they used for data collection. The number of interviewees was not mentioned. Remove the words "we entered."

Line 194: I don't find anything related to data analysis in this section.

Line 279: Better to explain what happened to "Xiao Liuqiu".

Line 323 and 350: Write "marine spatial planning" instead of "sea space planning".

Line 358: This sentence should be removed. This type of sentence should always be avoided in a scholarly article.

Line 385: Subsections 5.2 and 5.3 should be moved from Section 5 to Section 4, where they should be added as 4.4 and 4.5. Only the conclusion should be in Section 5.

Author Response

Reviewer’s Comments

Author Responses

Reviewer #4

The abstract of this manuscript is not well organized. The authors did not mention their methods of data collection. I would like to see a short description of the methodology, results, and recommendations

The abstract has been revised, and the research method is also added.

Line 39: This sentence is not clear.

This sentence has been revised and rewritten.

Line 85: Rewrite the sentence for better clarity.

This sentence has been revised and rewritten.

Line 97 – 98: What is "Barbados' Programmer of Action"?

"Barbados' Programmer of Action" is a typo, which has been revised in the text.

Line 99: The Barbados Programme of Action for the Sustainable Development of SIDS (BPOA), a 14-point programme that identifies priority areas and specific actions necessary for addressing the particular challenges SIDS faces. Therefore, correct the sentence.

The sentence has been revised.

Line 105: No reference.

References have been added.

Line 124: What is "local creation"?

"local creation" definition and description has been added to the text.

Line 170: Is this article only for the Chinese people? If not, change the texts on the map. The resolution of the map is also very poor. My suggestion is to use a different map. Unfortunately, the readers won't be able to find the exact location of the sampling site on this map.

The Figure 1 has been revised and follow reviewer suggestions.

Line 172: Section 3.2 needs a radical change. This section lack clarity and completeness. The authors should not mention "various" but rather mention the specific name of the materials they used for data collection. The number of interviewees was not mentioned. Remove the words "we entered."

Section 3.2 has been rewritten, and detailed material names, interviewees and number of people have been added. This sentence has been revised

Line 194: I don't find anything related to data analysis in this section.

The analysis has been supplemented in the text and compared with other island cases, and then the viewpoints of this paper are summarized.

Line 279: Better to explain what happened to "Xiao Liuqiu".

It was added to explain that the ecological environment of Liuqiu was destroyed because the tourists exceeded the carrying capacity.

Line 323 and 350: Write "marine spatial planning" instead of "sea space planning".

This "marine spatial planning" has been instead of "sea space planning"

Line 358: This sentence should be removed. This type of sentence should always be avoided in a scholarly article.

This sentence has been removed.

Line 385: Subsections 5.2 and 5.3 should be moved from Section 5 to Section 4, where they should be added as 4.4 and 4.5. Only the conclusion should be in Section 5.

I would like to thank the reviewer suggestion that moving 5.2 and 5.3 to Chapter 4 will help to simplify the chapter arrangement. However, since the above two chapters are policy and follow-up research suggestions, considering the coherence of the overall article, the original chapter arrangement is maintained. In addition, a more detailed supplementary explanation is given in Section 5.2.

Reviewer 5 Report

Dear editor,

The manuscript in view is of interest and well written.

My suggestion is to enrich the introduction and the discussion with other examples. Generally, some places around the globe face the same issues and prospects from the future. Comparisons should be made.

Author Response

Reviewer’s Comments

Author Responses

Reviewer #5

My suggestion is to enrich the introduction and the discussion with other examples. Generally, some places around the globe face the same issues and prospects from the future. Comparisons should be made.

The article has supplemented the experience of sustainable tourism development in Sipadan Island in Malaysia and Hananuma Bay in Hawaii and discussed in different cases.

Round 2

Reviewer 3 Report

Dear Authors,

Appreciate the revisions done for the manuscript based on the comments. However, some of the comments given in the manuscript itself were not addressed. Those were basically about the writing style and difficulties to easy understanding to the reader. Because some of those sentence are too long. However, I believe authors will take those pints positively.

Following sentence line 158 and 159 need to write in past tense.

We investigate the ecological environment, industrial economy, and human aspects of the Island, and take stock of existing local policies and practices

Thank you

Author Response

Thanks for the suggestion.

This sentence has been revised in past tense as follow:
We investigated the ecological environment, industrial economy, humanities and society aspects of the Island, and take stock of existing local policies and practices.

This manuscript is a resubmission of an earlier submission. The following is a list of the peer review reports and author responses from that submission.